# Advances in Plant GABA Research: Biological Functions, Synthesis Mechanisms and Regulatory Pathways

**DOI:** 10.3390/plants13202891

**Published:** 2024-10-15

**Authors:** Yixuan Hu, Xin Huang, Qinglai Xiao, Xuan Wu, Qi Tian, Wenyi Ma, Noman Shoaib, Yajie Liu, Hui Zhao, Zongyun Feng, Guowu Yu

**Affiliations:** 1State Key Laboratory of Crop Gene Exploration and Utilization in Southwest China, College of Agronomy, Sichuan Agricultural University, Chengdu 611130, China; 202200303@stu.sicau.edu.cn (Y.H.); 202200322@stu.sicau.edu.cn (X.H.); 202200313@stu.sicau.edu.cn (Q.X.); 202200312@stu.sicau.edu.cn (X.W.); 202200328@stu.sicau.edu.cn (Q.T.); 202200349@stu.sicau.edu.cn (W.M.); zhaohui_hbu@126.com (H.Z.); 2Chengdu Institute of Biology, Chinese Academy of Sciences, Chengdu 610041, China; noman@cib.ac.cn

**Keywords:** γ-aminobutyric acid, GABA shunt, polyamine degradation pathway, glutamic acid decarboxylase, abiotic stress

## Abstract

The γ-aminobutyric acid (GABA) is a widely distributed neurotransmitter in living organisms, known for its inhibitory role in animals. GABA exerts calming effects on the mind, lowers blood pressure in animals, and enhances stress resistance during the growth and development of plants. Enhancing GABA content in plants has become a focal point of current research. In plants, GABA is synthesized through two metabolic pathways, the GABA shunt and the polyamine degradation pathway, with the GABA shunt being the primary route. Extensive studies have investigated the regulatory mechanisms governing GABA synthesis. At the genetic level, GABA production and degradation can be modulated by gene overexpression, signaling molecule-induced expression, transcription factor regulation, and RNA interference. Additionally, at the level of transporter proteins, increased activity of GABA transporters and proline transporters enhances the transport of glutamate and GABA. The activity of glutamate decarboxylase, a key enzyme in GABA synthesis, along with various external factors, also influences GABA synthesis. This paper summarizes the biological functions, metabolic pathways, and regulatory mechanisms of GABA, providing a theoretical foundation for further research on GABA in plants.

## 1. Introduction

### 1.1. Physical and Chemical Properties

γ-aminobutyric acid (GABA) is a four-carbon non-protein amino acid with the molecular formula NH_2_(CH_2_)_3_COOH, which was first discovered in potato tubers over 70 years ago [1,2]. GABA is a white, flaky, or needle-like crystal that is slightly soluble in hot ethanol, insoluble in most common organic solvents, and highly soluble in water. It is deliquescent and dissociates in aqueous solutions, typically existing as a zwitterion. GABA has a molecular weight of 103.12, an isoelectric point (pI) of 7.19, and a decomposition temperature of 202 °C. It is widely distributed in plants and animals, occurring in the seeds, rhizomes, and tissue fluids of legumes and herbs, while in animals, it is almost exclusively found in nervous tissue.

### 1.2. Biological Functions of GABA in Plants

GABA can act as a signaling molecule directly involved in the regulation of biotic and abiotic stresses. GABA triggers a defense response in plants when they experience biotic stress, particularly from insect attacks. Research has shown a positive correlation between increased GABA levels and enhanced resistance to such stress. For instance, plants with higher GABA content have demonstrated greater resistance than their wild-type counterparts [3]. Moreover, GABA inhibits neuronal transmission in insects, thereby serving as a defensive mechanism [4]. Additionally, GABA accumulation has been found to mitigate the toxic effects of certain compounds produced during insect feeding [5]. These findings indicate that GABA is crucial in reducing plant damage caused by biotic stress. GABA is a signaling molecule in plants able to enhance plant adaptation to abiotic stresses. A number of studies have shown that GABA can enhance plant resistance to hypoxia [6], salinity, drought [7], high temperature [8], and heavy metal stress [9]. In addition, specific concentrations of GABA can regulate the antioxidant system of plant growth, thus reducing oxidative damage in plants [10], GABA and proline in barley and tobacco leaves can increase their salt tolerance [11], GABA may be as a non-toxic permeability clear free radicals (ROS) produced under salt stress [12], which is consistent with the previous conclusion that GABA can inhibit the production of hydrogen peroxide under salt stress [13], GABA treatment, the germination rate of wheat and maize under salt stress is significantly increased [14]. In addition, experiments on morning glory pepper seeds under salt stress revealed that treatment with 6.0 µmol/L GABA improved both germination ability and salt tolerance [15]. In another study on rice seedlings, GABA treatment resulted in increasing plant height and leaf area, enhancing root strength and salt tolerance [16]. Additionally, different concentrations of GABA have been shown to alleviate the negative effects of drought stress on maize [17]. Research on durum wheat has further indicated that under drought conditions, GABA shunt metabolism is activated, which helps maintain carbon and nitrogen balance, regulate amino acid metabolism, and support plant growth [18].

In addition to its direct involvement as a signaling molecule in the regulation of the physiological state of plants, GABA can also interact with various plant hormones such as ABA, GA, auxin, CTK, and ethylene to help plants cope with abiotic stress [19]. For example, under salt stress, GABA application to *Caragana intermedia* and Poplar can promote the production of ABA and hydrogen peroxide while also regulating the production of ACS, ACO and ethylene [13]. Under drought stress, GABA can not only affect ALMT 9 channels but also promote the synthesis of PAs and inhibit the catabolism of PAs, which can greatly increase the content of different types of PAs in response to drought [20]. Conversely, addition of ABA and auxin under adversity could also increase GABA production [21]. In the face of stress and stress-related response and metabolism, The crosstalk between auxin and other phytohormones can all cause signaling interference from GABA, Overall, GABA and multiple phytohormones are mutually influencing the interacting [22], GABA increases the ethylene content by increasing the expression of genes such as ACC synthetase and ACC oxidase (ACO), Whereas, when the plant CK is deficient, The amount of LEA genes and glyoxylate reductase (GLYR) also decreases (GLYR is the enzyme involved in GABA catabolism) [23].

GABA also functions as a signaling molecule in plants, where it plays a key role in regulating stomatal movement. It has been observed that GABA inhibits ion channels in stomatal guard cells, thereby controlling stomatal opening, water loss, and drought tolerance [24]. In studies involving tomato plants, GABA was found to reduce stomatal conductance and diameter, promoting stomatal closure [25]. In *Arabidopsis*, guard cells produce GABA when reduced water, and GABA suppresses stomatal opening, a negative regulation of [26]. In addition to *Arabidopsis* and other plants also have this regulation, the reason is mainly with water, GABA increased, this signal through anion channel ALMT 9 transduction, and anion channel ALMT9 is the process of the main route [27], to reduce the stomatal opening, also reduce the transpiration water loss, improve the water use efficiency (WUE), enhance the drought resistance. Interestingly, GABA helped plants resist stress under both single and compound stress, but the relative GAD expression of compound stress was not consistent with stomatal movement compared to single stress. Balfagon et al., exposed to high light intensity (HL), heat stress (HS) and its combination (HL + HS), found increased transpiration and stomatal conductivity, which may be related to reduced water loss, lower leaf temperature and the limited photosynthesis rate of *Arabidopsis* itself [28]. Further research suggests that GABA may also be involved in intercellular signaling within stomata [29]. Additionally, GABA signaling is thought to regulate ion fluxes across the protective membranes of stomata, although this requires further investigation [30]. These findings underline the importance of GABA as a signaling molecule in stomatal regulation [31].

Furthermore, GABA has been shown to influence seed germination, enhancing germination rates under certain concentrations. Experiments with licorice seeds treated with varying concentrations of GABA demonstrated that a specific GABA concentration increased the content of seed respiratory metabolites and promoted the tricarboxylic acid cycle, both of which are beneficial for seed germination [32]. Similarly, treatment of soybean seeds with GABA led to an increase in germination potential and vigor index, likely due to a reduction in abscisic acid (ABA) content, an increase in ethylene levels, and the promotion of seed germination regulation by endogenous hormones [33]. Another study found that wheat seeds germinated more quickly under certain treatments, likely due to enhanced GABA shunt metabolism [34], indicating that the careful application of GABA can effectively promote seed germination.

## 2. Transporters of GABA

A variety of GABA transporters have been found on plant cell membranes and organelles. These transporters mainly include aluminum-activated malate transporters (ALMTs), GABA transporters (GATs), cationic amino acid transporters (CATs), AAP3, ProTs and bidirectional amino acid transporters (BATs). Transporters located in the cell membrane are ALMTs, GATs, AAP3 and ProTs, in addition, ALMTs appear to be recognized as a specific GABA receptor on the surface of plant cell membranes [35]. Transporters located in the organelle membrane are CATs and BATs. GABA is controlled by shuttling across cell membranes and organelles, entering and exiting cells for metabolism [36,37,38]. The following is an introduction to the functions and mechanisms of these transporters.

### 2.1. Transporters on Cell Membranes

#### 2.1.1. ALMTs

Aluminum-activated malate transporters (ALMTs) are bidirectional transmembrane anion transporters [39], 12 ALMT family genes have been found in plants [40], and ALMT1 has been found in rice, wheat, rape, *Arabidopsis* and other plants. In 2018, Ramesh found that ALMT1 on plant cell membranes can efficiently transport GABA [39], and in addition, ALMTs respond to various signals, are activated by anions, and negatively regulate GABA [35].The specific transport mechanism of ALMTs is as follows: Previous studies on GABA and malate have shown that anions can activate ALMT1 [41], and H^+^-ATPase can input and export amino acids to cells through the protons produced by the plasma membrane, resulting in a potential difference between inside and outside the membrane. When the intracellular pH is low, aluminum ions promote efflux of GABA through ALMT1. When the extracellular are acidified, GABA was transfered into inside of cell through ALMT1. These studies have shown that pH can affect the potential difference between inside and outside the membrane, and then affect the direction of ALMT1’s transport of GABA [42,43,44]. At present, ALMT family genes have been cloned and identified successively, and their protein sequences and transport mechanisms have been analyzed, and ALMT1 is still the most studied. It is generally confirmed that when GABA content increases, the activity of ALMT decreases, that is, GABA negatively regulates the activity of ALMT [42]. Yu Long’s study proved that GABA inhibits the transport of anions in wheat by changing the active structure of ALMT1, and this conformational change is similar to the conformational transformation of aspartate aminotransferase (AAT) [45]. At present, there are few studies on regulating the activity of ALMTs to regulate the transport of GABA, and most of them remain in the exploration of the transport mechanism. However, the interaction between GABA and ALMT can be used as a signal molecule within the implant to regulate the transmembrane transport of GABA.

#### 2.1.2. GATs

GATs is a class of membrane transmembrane transporter protein, GAT gene belongs to AAAP gene family, and four GAT homologous genes (GAT1, GAT2, GAT3, GAT4) have been found in plants [36,46]. At present, GAT1, which is located on the cell membrane, can transport GABA across the membrane and flow between the apoplast and the cytoplasm. Both GAT1 and ALTM1 are transmembrane transporters of GABA, but it is worth noting that Al^3+^ blocks the influx of GABA from apoplast to the cytoplasm via ALMT1, but it has no effect on GAT1 [42]. Up to now, GAT1 has been cloned in potato, *Arabidopsis*, rice and other crops, and related studies have been carried out in *Arabidopsis*. AtGAT1 in *Arabidopsis thaliana* transports GABA across membranes through proton coupling, which is mainly driven by the proton electrochemical potential gradient and the high affinity of AtGAT1 for GABA [46].

#### 2.1.3. AAP3 and ProTs

At present, two GABA transporters (AAP3 and ProT2) on the cell membrane have been identified by heterologous recombination in yeast, but the affinity of these two proteins to GABA is relatively low [47,48,49]. Regarding the transport function of OsProTs family proteins in rice, it has been reported that proline transporters not only have a transport function for Pro, but also have a transport function for GABA. OsProT2 is heterologous expressed in Xenopus oocytes. It is found that OsProT2 is a proton cotransporter and has specific transport function for L-Pro, but whether it transports GABA remains to be studied [50,51]. In *Saccharomyces cerevisiae* mutants with amino acid transport defects, OsProT1 and OsProT3 can specifically absorb Pro and GABA, and OsProT3 has a stronger transport capacity for Pro than OsProT1, while OsProT2 has no transport properties for various amino acids. This may be due to the inability of OsProT2 to localize to the plasma membrane in yeast [52]. In summary, ProTs and AAP3 can transport GABA, but the affinity is not strong, and its related causes and regulatory factors need to be further studied.

### 2.2. Transporters on Organelle Membranes

#### 2.2.1. CAT9

CATs is a cationic transporter located on the vacuole membrane and belongs to the APC gene family [53,54,55]. At present, nine CAT genes have been found in plants, among which the transporter encoded by CAT9 is mainly responsible for the bidirectional transport of GABA between vacuole and cytoplasm. At present, CAT9 has been found in tomato, potato, *Arabidopsis*, rice and other crops [53,55], and the transport dynamic mechanism of CAT9 has been verified in tomato. SICAT9 transport in tomato mainly has two driving forces: one is the proton driving force of chloroplast proton pump, and the other is the chemical potential difference caused by the concentration gradient of substrate [56]. Interestingly, in addition to GABA, SlCAT9 also transports protein amino acids such as glutamic acid and aspartic acid, but the transport conditions are relatively strict. Glutamic acid is the substrate for GABA synthesis, and aspartic acid can also be converted into glutamic acid, so it is speculated that SlCAT9 co-regulates GABA transformation with these amino acids [57]. Studying the co-transport mechanism and regulatory factors of CAT9 will be a promising direction, which can effectively regulate GABA transport. Relevant studies have shown that some external factors may affect the expression of CATs, such as supplemental nitrogen sources and stress. For example, when tea leaves are fed with different nitrogen sources or under stress conditions, the expression of CsCATs will be inhibited or induced, thus inhibiting or promoting the transport of GABA. The content of glutamate will increase under nitrogen treatment or stress conditions, which will also promote the synthesis of GABA [58]. AtCAT9 in *Arabidopsis thaliana* can directly or indirectly balance intracellular amino acid concentration in the absence of leaf nitrogen [59]. In addition, the expression level of CATs is also different in different parts, and the expression level of AtCAT9 in *Arabidopsis thaliana* is relatively high in root, stem, leaf and flower tissues, indicating that GABA transport is relatively active in these parts [59].

#### 2.2.2. BAT1

BAT is a bidirectional transmembrane protein on mitochondrial membrane. Seven homologous genes have been identified in plants, belonging to the APC gene family [60,61], in which the transporter encoded by BAT1 can transport amino acids. AtGABP in *Arabidopsis thaliana* is a splice variant of AtBAT1 that can transport GABA [62]. Unlike AAP3 and ProT2, AtGABP cannot transport proline, and the co-expression of AtGABP gene is highly correlated with the expression of *SSADH* gene. It can be speculated that AtGABP is associated with some reactions of GABA metabolism, and may jointly participate in the regulation of GABA shingling and TCA cycle [62]. It is worth noting that AtBAT1 in *Arabidopsis* can transport a variety of amino acids such as arginine and lysine, but has no transport activity for GABA [37]. At present, there is relatively little research on *BAT* genes, and the transport mechanism of BAT transporter and its splicing variant GABP will be a hot direction.

## 3. Metabolism of GABA in Plants

There are two primary pathways for GABA synthesis in plants: the GABA shunt and the polyamine degradation pathway. The GABA shunt is the predominant pathway for GABA synthesis in most plants, while the polyamine degradation pathway remains less explored. GABA synthesis has been extensively studied in plants such as tomatoes [63], rice [64], watermelon [65], quinoa [66], wheat [67], and corn [68] but research on barley remains limited.

### 3.1. GABA Shunt

The GABA shunt, also known as the GABA bypass, is the major pathway for GABA synthesis in most higher plants [69]. In the cytoplasm, where there is a high concentration of H^+^ ions, glutamic acid (Glu) undergoes an irreversible reaction catalyzed by the enzyme cytosolic glutamate decarboxylase (GAD), leading to the production of GABA.

There are two main pathways for glutamate synthesis in plants (Figure 1). The first pathway involves glucose undergoing glycolysis, Under the catalysis of pyruvate dehydrogenase system, pyruvate generates acetyl-Coenzyme A in the mitochondrial matrix, and then enters the tricarboxylic acid cycle (Kreb’s cycle, also known as TCA). The citrate in TCA passes under the catalysis of a series of enzymes to generate the intermediate product α-ketoglutaric acid. This α-ketoglutaric acid can then be catalyzed by glutamate dehydrogenase (GDH). Glutamate dehydrogenase (GDH) can be divided into three types according to their coenzymes: NADH-dependent, NADPH-dependent and NADH/NADPH-dependent [70], NADH-dependent GDH is usually involved in glutamate catabolism, while NADPH-dependent GDH requires nitrogen assimilation, and dual-dependent GDH is usually in mammals [71]. The second pathway involves the plant’s response to an excess of free ammonia within the cell, which can be toxic. In this case, glutamine synthetase (GS) catalyzes the reaction of free ammonia with glutamate to form glutamine. This glutamine is then catalyzed by glutamate synthetase (GOGAT) in the presence of α-ketoglutarate, regenerating two molecules of glutamate. Glutamate synthase (GOGAT) in plants can also be divided into two types of coenzyme: Fd-GOGAT and NADH-GOGAT, the former mainly exists in green tissues such as chloroplast and plastid, while the latter mainly exists in non-green tissues such as nodules and stems, participating in nitrogen fixation, and the specific site of the reaction is in the mitochondria [72].The GS/GOGAT cycle is the primary pathway for the assimilation of ammonia and the synthesis of amino acids in plants [73].

After GABA formation by glutamate under the glutamate decarboxylase, the generated GABA is transferred from the cytosol to the mitochondria through the BAT1. GABA and α-ketoglutarate are catalyzed to glutamate (Glu) and succinate semialdehyde (SSA) by GABA transaminase (GABA-TK). Succinate semialdehyde is catalyzed to succinate by succinate semialdehyde dehydrogenase (SSADH), and succinate eventually entering the TCA cycle to be degraded (Tricarboxylic acid, TCA) [21].

### 3.2. Polyamine Degradation Pathway

The polyamine degradation pathway, also known as the putrescine degradation pathway, is primarily responsible for GABA synthesis in legumes, with less activity observed in grasses [74]. Additionally, it has been found that this pathway contributes to GABA synthesis in plants under stress conditions [75]. Polyamines (PAs) in plants include putrescine (Put), spermine (Spm), and spermidine (Spd). Putrescine is mainly generated through the decarboxylation of ornithine, although it can also result from spermidine decarboxylation. Once formed, putrescine is further catalyzed by diamine oxidase (DAO) to produce 4-aminobutanal, which is then transformed into GABA by aminobutyraldehyde dehydrogenase (AMADH). The synthesized GABA can enter the tricarboxylic acid cycle (TCA cycle) through the catalytic actions of GABA transaminase (GABA-T) and succinic semialdehyde dehydrogenase (SSADH) [76]. Putrescine can also be converted into spermidine and spermine through reactions catalyzed by spermidine synthase and spermine synthase respectively. Spermidine, in turn, can be catalyzed by polyamine oxidase (PAO) to produce 4-aminobutyraldehyde, which further produces GABA [74].

## 4. GABA Regulation

Based on the synthesis and metabolic pathways of GABA, its regulation can be categorized into three main aspects: gene-level regulation (Figure 2A), regulation of key enzyme activities (Figure 2A), and regulation by external factors (Figure 2B).

### 4.1. Gene Level Regulation

*GAD* (glutamate decarboxylase) genes have been identified in a variety of plants (Table 1). Alterations in the structure of some genes, as well as the regulation of transcription factors, influence GABA production. In *Arabidopsis*, there are five GAD family genes: *GAD1*, *GAD2*, *GAD3*, *GAD4*, and *GAD5* [77]. These genes exhibit differential expression across various parts of the plant. For instance, *GAD1* is highly expressed in roots, *GAD2* shows very high expression in seedlings after two weeks and in carpels, buds, wreaths, roots, and stems, while *GAD4* is predominantly expressed in pollen tubes. *GAD3* and *GAD5* have lower expression levels in the same tissues [78,79,80]. A common feature of these *GAD* family genes is the presence of CAAT-box, TATA-box, and MYC elements in their promoters and they could be binded by transcription factors to regulate the expression of *GAD* genes [81]. Myelocytomatosis oncogenes (MYC) transcription factors (TFs) belong to the bHLH (basic helix-loop-helix) family, which play a central role in plant growth, development, adaptation to biotic and abiotic stress, as well as secondary metabolism [82,83,84,85]. At present, They have been mainly studied in rice, wheat and Arabidopsis [86].

In addition, MYB transcription factors were involved in regulating various stresses in plant life cycle, and OsMYB55 belongs to one of the MYB transcription factors, which can promote amino acid metabolism during plant growth under high temperature to improve rice yield [87]. In rice, the transcription factor OsMYB55 binds to the promoter regions of target genes, directly activating the expression of several genes, including glutamine synthetase (Os*GS1*, Os*GS2*), glutamine amidotransferase (*GAT1*), and glutamate decarboxylase 3 (*GAD3*) [87]. Overexpression of *OsMYB55* in rice lines resulted in increased GABA content in leaves, especially under high-temperature conditions, with a more obvious effect in transgenic plants compared to wild-type plants. Additionally, total amino acid content increased following the overexpression of *OsMYB55*. Transcriptome analysis identified Os*GS2*, *GAT1*, and *GAD3* as potential targets of *OsMYB55*. This transcription factor has been shown to bind to the promoters of these genes to activate them [87].

In bananas, exogenous ethylene was found to significantly induce the expression of Ma*GAD1*, with regulation primarily occurring at the transcriptional level. Research in *Arabidopsis* revealed that Ma*GAD1* enhances the exogenous ethylene sensing capacity of the upstream components of the ethylene signaling pathway, particularly through the regulation of *ACS4*, a member of the ACC synthase family involved in ethylene biosynthesis [88]. This regulation impacts the endogenous ethylene biosynthesis pathway in *Arabidopsis*. In poplar, members of the *GAD* gene family may be induced by anaerobiosis, low temperature, and drought, as their regulatory elements correspond to these conditions. Poplar also contains two GABA-T family genes with elements related to light response, anaerobic induction, and abscisic acid. Additionally, *PopGABA-T1* includes elements associated with meristem organization and cyclic regulation. Poplar has two SSADH gene families, both of which include elements related to light and gibberellin responses, and *PopSSADH1* also has elements involved in anaerobic induction, defense, and stress response [89]. By regulating these elements and the local environment of the plant, we can regulate GABA metabolism and promote the synthesis of GABA to increase its content.

Inhibiting the expression of *SSADH* and *GABA-T* genes is an effective approach to increase GABA content in plants by reducing GABA degradation (Figure 2A). For instance, RNA interference technology has been employed to construct an *SSADH* gene RNA interference strain (MA-i SSADH), leading to a significant decrease in *SSADH* gene transcript levels compared to the prototrophic strain. In rice, an RNA interference vector targeting the GABA transaminase 1 (*OsGABA-T1*) gene was developed and successfully transformed into the japonica rice variety “Ning Japonica 1” using Agrobacterium-mediated transformation. This method effectively suppressed the expression of *OsGABA-T1* and its related gene *OsGABA-T2* [90]. In conclusion, the expression of *GABA-T* and *SSADH* was inhibited by RNA interference, thus inhibiting GABA degradation and increasing its content.

### 4.2. Effect of GAD Enzyme Activity on GABA Synthesis

GABA production in plants and microorganisms primarily occurs through the GABA shunt, a pathway catalyzed by GAD. It is the key enzyme in this pathway, and its activity can be enhanced by optimizing various factors, including temperature, pH, Ca^2+^/CaM, the coenzyme pyridoxal phosphate (PLP), and metal ions (Figure 2A).

#### 4.2.1. Temperature Regulation of GAD

The optimal temperature for GAD activity varies significantly across different species, generally ranging from 30 °C to 50 °C in plants and microorganisms [91]. For example, the optimal temperature for GAD in bananas during ripening is 37 °C, with GAD activity decreasing by 23% at 25 °C and by 39% at 55 °C [92]. In potatoes, the optimal temperature for GAD activity is also 37 °C [93]. In lactic acid bacteria, 52 °C is identified as the optimal temperature for GAD activity [94]. Meanwhile, in pumpkin, the optimal reaction temperature for GAD is 30–35 °C; however, the enzyme is heat-sensitive and becomes inactive after short exposure to temperatures above 50 °C [95]. These findings indicate that the optimal conditions for GAD activity vary widely among species, suggesting that adjusting the reaction temperature can significantly improve GAD activity.

#### 4.2.2. The pH Regulation of GAD

The optimal pH for GAD activity varies depending on its source. Typically, GAD from microbial origins operates best within a pH range of 3.8 to 5.5 [96]. For example, the optimal pH for GAD in *E. coli* is between 3.8 and 4.5 [97], while in *Lactobacillus* it is between 4.0 and 5.0 [98], and in *Bacillus megaterium*, it is 4.5 to 5.0 [99]. For plant-derived GAD, the optimal pH is generally weakly acidic, mainly between 5.5 and 6.0. For instance, the optimal pH in soybeans is 5.9. Although soybean GAD can retain some activity between pH 5.0 and 8.0, its activity decreases sharply at pH levels above 8.0, with an 83% reduction at pH 9.0 [100]. Similarly, the optimal pH for GAD in corn embryos is 5.7, with a dramatic decline in activity at pH levels below 4.0 or above 8.0, showing an 80% decrease at pH 9.0 [101]. In rice embryos, the optimal pH for GAD is between 5.5 and 5.8 [102]. These studies demonstrate that pH significantly influences GAD activity, and adjusting the pH of the enzyme reaction can enhance GAD activity, thereby increasing GABA content in plants.

#### 4.2.3. Pyridoxal Phosphate of GAD

The PLP is a key coenzyme for GAD, enhancing GABA synthesis by promoting the decarboxylation of glutamate. Research has shown that GAD from barley embryo extracts, with a Km value of 22 mmol/L for L-glutamate, is activated 3.5-fold in the presence of pyridoxal phosphate. Further studies revealed that GAD in barley embryos exists in two molecular weight forms: a low molecular weight form, which is relatively inactive, and a high molecular weight form, which becomes more active upon storage. The presence of 2-mercaptoethanol, a thiol reagent, shifts the distribution of enzyme activity toward the low molecular weight form. However, once 2-mercaptoethanol is removed, the enzyme spontaneously binds to the high molecular weight form, increasing its activity. Additionally, the presence of oxygen in the extraction buffer promotes the formation of the higher molecular weight form, enhancing enzyme activity. In contrast, GAD from 3-day-old barley root extracts had a Km value of 3.1 mmol/L for L-glutamate and exhibited a 10% increase in activity with the addition of pyridoxal phosphate [103]. These findings suggest that pyridoxal phosphate can significantly boost GAD activity in plants, and supplementing this coenzyme in enzyme reactions can enhance both GAD activity and GABA production.

#### 4.2.4. Metal Ion Regulation of GAD

GAD activity is also influenced by various metal ions. In maize embryos, metal ions such as Mg^2+^, Mn^2+^, Cu^2+^, Al^3^^+^, Ag^+^, and Zn^2+^ do not significantly affect GAD activity at a concentration of 2 mmol/L. However, Ca^2+^ has a notable impact. When the Ca^2+^ concentration is below 400 μmol/L, there is a significant activation effect, with the most pronounced increase in GAD activity (31%) observed at 400 μmol/L compared to untreated groups [101]. Similarly, in soybean, Mg^2+^ at 2 mmol/L has little effect, resulting in only a 3.5% decrease in enzyme activity. KCl causes a slight decrease of about 7%, while KI and Ag^+^ have a more substantial inhibitory effect, reducing enzyme activity by approximately 13% [100]. In rice bran, Mg^2+^, Mn^2+^, and Al^3^^+^ do not significantly affect enzyme activity, while KCl has a slight effect. However, KI and Ag^+^ significantly inhibit enzyme activity [104]. These findings suggest that certain metal ions, particularly Ag^+^ and KI, consistently inhibit GAD activity across various plant species. Reducing the presence of these ions could help prevent the inhibition of GAD activity and enhance GABA production in plants.

#### 4.2.5. Ca^2+^/CaM Regulation

GAD in plants is a calmodulin (CaM)-binding protein and its activity is regulated by the Ca^2+^/CaM complex. The calmodulin-binding region is located at the C-terminal end of GAD, where Ca^2+^/CaM acts as an activator, significantly increasing GAD activity [105]. Truncating the C-terminus of rice OsGAD2 results in an enzyme with higher activity than the wild-type at any pH, with over 40-fold higher activity at physiological pH [105]. In mung beans, calmodulin significantly enhances the relative enzyme activity of GAD at certain concentrations [106]. Studies in Petunia show that the Ca^2+^/CaM complex must bind to the calmodulin-binding domain of GAD to form a complex that promotes GABA synthesis. Additionally, in a pea and buckwheat enzyme solution, the amount of GABA synthesis increases as Ca^2+^ concentrations rise from 0 to 0.8 mmol/L, reaching the highest GABA production at 0.8 mmol/L. Beyond this concentration, from 0.8 to 1.0 mmol/L, GABA synthesis decreases, possibly due to the inhibitory effects of excessively high Ca^2+^ levels on GAD activity [107]. These findings indicate that by optimizing the concentration of Ca^2+^/CaM during GAD enzyme reactions, it is possible to enhance GAD activity and subsequently increase GABA production.

## 5. External Factors Regulating GABA

### 5.1. Temperature Treatment

Extreme temperature treatment can increase the content of GABA in plants (Figure 2B). Temperature stress in plants is categorized into high-temperature stress and low-temperature stress. Low-temperature stress includes both cold damage and freeze damage. Freeze damage occurs when ice forms within plant tissues at temperatures below 0 °C, leading to the formation of large ice crystals that severely damage the cell membrane system. This damage allows Ca^2+^, H^+^, and other ions to penetrate deeper into the cytoplasm, where they bind to the Ca^2+^/CaM binding domain in glutamic acid decarboxylase (GAD). This binding increases the expression of calmodulin, which activates GAD to catalyze the conversion of L-glutamate to GABA. The disruption of the cellular membrane system also facilitates the exchange of substrates between cells, allowing more L-glutamate to bind with GAD, resulting in increased GABA accumulation [108]. For example, low-temperature treatment of germinated maize kernels at −18 °C significantly increased GABA content, peaking at 1.38 mg/g—2.3 times higher than in untreated kernels. In contrast, maize kernels exposed to 5 °C without frost damage showed the least increase in GABA content, with only a 0.91-fold increase compared to untreated kernels [109]. High-temperature stress occurs when ambient temperatures exceed the optimal range for plant growth, leading to physiological and biochemical disruptions. This stress can be categorized into short-term heat stress and long-term desiccation. High temperatures cause an accumulation of reactive oxygen species (ROS) within cells, increased membrane lipid peroxidation, and electrolyte leakage, all of which impair photosynthesis and alter antioxidant enzyme activity [110]. Numerous studies have demonstrated that GABA enhances plant stress tolerance by stabilizing cell membrane structures, regulating cytoplasmic pH, inducing ethylene production, reducing ROS damage, and promoting the synthesis of biomolecules [111]. For instance, under prolonged drying treatment at 42 °C, the GABA content in maize seedlings significantly increased, peaking at 48 h before rapidly declining [112]. Similarly, in tea plants exposed to day/night temperatures of 42 °C/40 °C, endogenous GABA levels rose significantly, increasing by 75.56%, 88.99%, and 60.13% at 4, 8, and 24 h, respectively, before also decreasing [2].

### 5.2. Other Treatments

It has been found that in the pre-germination period of brown rice, with the help of ultrasonic treatment, the use of soaking germination can increase the GABA content in brown rice (Figure 2B). With the prolongation of germination time, GAD activity had a tendency to increase, then decrease and then increase [113]. Similarly, the combination of soaking buckwheat in EFW (electric field water) with high-voltage electric field (HVEF) treatment has been shown to significantly increase GABA accumulation during germination (Figure 2B). At 24 h, the combined treatment reached a peak GABA content of 198.72 ± 0.75 mg/100 g, showing a significant difference (*p* < 0.05) compared to other treatment groups [114]. This suggests that the combined use of EFW soaking and HVEF treatment is more effective at enhancing GABA content in sweet buckwheat than either treatment alone. Pulsed electric field (PEF) treatment is another method that influences membrane permeability, enhancing the interaction between substrates and enzymes, and thereby promoting GABA synthesis through electroporation (Figure 2B). The effectiveness of PEF treatment depends on carefully designed factors, including pulse intensity, pulse duration, and pulse type [115].

## 6. Summary and Outlook

GABA plays an important role in the resistance to biotic and abiotic stresses. The main GABA transporters on the cell membrane are ALMTs and GATs, among which ALMTs is also considered to be GABA receptors. CAT9 is a cationic transporter, which can also participate in GABA transport. BAT1 is a bidirectional GABA transporter on mitochondria, and all the above transporters can participate in GABA transport. The metabolism of GABA in plants primarily occurs through the GABA shunt and the polyamine degradation pathway. Regulation of key enzymes within these pathways is likely crucial for increasing GABA content in plants, with the GABA shunt being the dominant metabolic route in most plants and the primary target for regulation. Enhancing GABA in plants can be approached in several ways. First, the key enzyme genes involved in GABA synthesis can be regulated through gene overexpression, transcription factor manipulation, or RNA interference technology to enhance the expression of GAD or inhibit the expression of GABA-T. Second, identifying and optimizing the conditions that maximize GAD enzyme activity can further promote GABA synthesis. Additionally, external factors such as extreme temperature treatment, ultrasonic treatment, and other innovative methods have been extensively studied and can be used to regulate GABA levels. Moreover, regulating glutamate synthesis by targeting key enzymes in nitrogen metabolism, such as GOGAT and GS, may increase glutamate content, thereby providing more substrate for the GABA shunt and promoting GABA synthesis. Although GAD genes are part of multigene families, further research is required to understand the forms of GAD present in plants, as well as the gene interactions and localization of the main effector genes within these families. In particular, research on the GAD gene family in crops like barley, to identify optimal conditions for GAD enzyme activity, represents a substantial direction for future studies.

## Figures and Tables

**Figure 1 plants-13-02891-f001:**
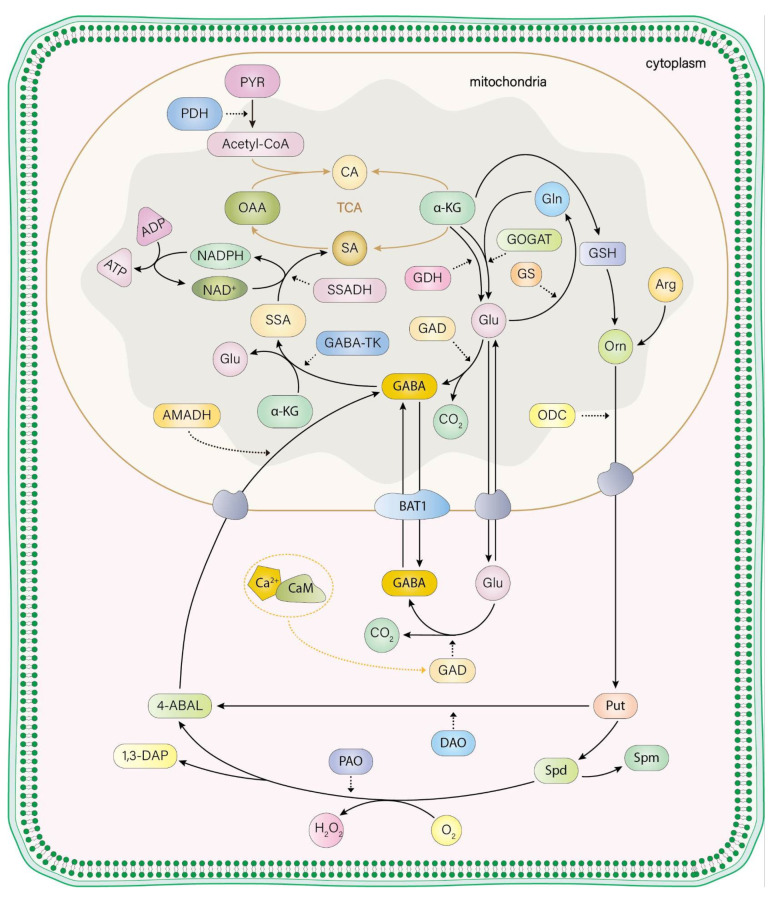
Metabolism of GABA in plants, PDH: Pyruvate Dehydrogenase, Acetyl-CoA: Acetyl Coenzyme A, CA: Citric Acid, OAA: Oxaloacetic Acid, α-KG: α-Ketoglutaric Acid, SA: Succinic Acid, SSA: Succinate Semialdehyde, NADPH: Nicotinamide Adenine Dinucleotide Phosphate, NAD^+^: Nicotinamide Adenine Dinucleotide, SSADH: Succinate Semialdehyde Dehydrogenase, Glu: Glutamic Acid, Gln: Glutamine, GS: Glutamine Synthase, GOGAT: Glutamate Synthetase, GDH: Glutamate Dehydrogenase, GSH: Glutathione, Orn: Ornithine, Arg: Arginine, ODC: Ornithine Decarboxylase, AMADH: Amino Acid Dehydrogenase, Put: Putrescine, Spm: Spermine, Spd: Spermidine, PAO: Polyamineoxidase, DAO: Diamine Oxidase, CaM: Calmodulin, GAD: Glutamate Decarboxylase, GABA: γ-aminobutyric Acid, BAT1: GABA Bidirectional Transmembrane Protein, GABA-TK: GABA-transaminase, 4-ABAL: 4-aminobutanal, 1,3-DAP: 1,3-diaminopropane.

**Figure 2 plants-13-02891-f002:**
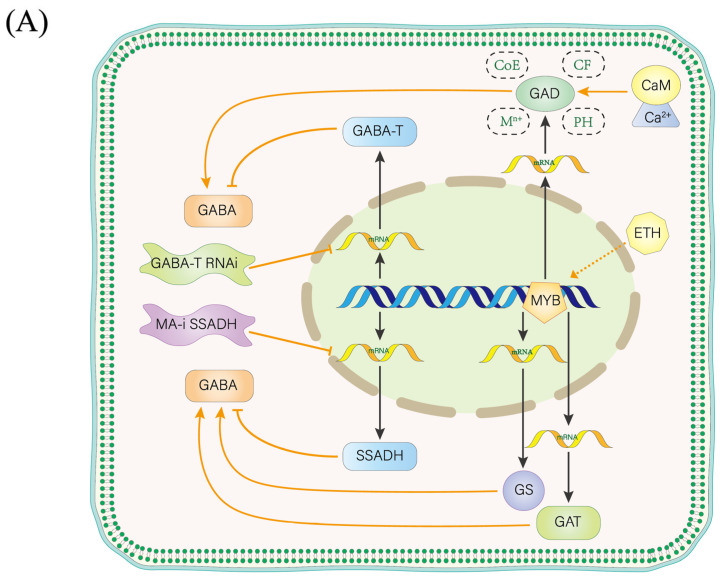
(**A**) Regulation of GABA at the gene and protein levels; (**B**) Regulation of GABA by physical factors. GABA: γ-aminobutyric acid, GAT: Transferase of glutaminase, GATs: Transferase of glutaminase family, ProTs: Proline transferase family, Temp: Temperature, GABA-T: gamma-aminobutyric acid transferase, Pro: proline GS: Transferase of glutaminase, SSADH: succinate semi-aldehyde dehydrogenase, GABA-T IFN: GABA-T interferon, MA-iSSADH: SSADH gene RNA interference Strain, M^n+^: Metal ion, CaM: calmodulin, Ca^2+^: Calcium ion, GAD: glutamate decarboxylase, PH: pH, CoE: Coenzyme CF: Cogroup, ETH: ethylene, MYB: A kind of transcription factor in plants.

**Table 1 plants-13-02891-t001:** Number of GAD genes in different plants.

Assortment	Numbers of GADs	Authors of the Reference	Years of Publication
*Arabidopsis*	5	Shelp, B.J., et al.	2017 [77]
*Citrullus lanatus*	3	Li, M., et al.	2023 [65]
*Oryza sativa Janica*	5	Zhou, L.	2015 [64]
*Solanum lycopersicum*	5	Sun, X., et al.	2022 [63]
*Chenopodium quinoa wild*	8	Li, L.	2022 [66]
*Zea mays* L.	5	Wang, Y.	2023 [68]

## Data Availability

Not applicable.

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
