# Peer review of "Advances in Plant GABA Research: Biological Functions, Synthesis Mechanisms and Regulatory Pathways"

_plants, 2024, doi:10.3390/plants13202891_

Round 1
Reviewer 1 Report
Comments and Suggestions for Authors
This manuscript provided a detailed overview of the biological functions, synthesis mechanisms, and regulatory pathways of GABA, as well as a detailed summary of related research on GABA in plants. Finally, key enzyme genes controlling GABA biosynthesis, regulating transporters involved in GABA metabolism, and enhancing GAD activity were proposed to promote GABA synthesis in plants. Overall, this is a comprehensive study that provides theoretical basis and research direction for crop nutritional quality and resistance breeding. However, my points should be well addressed below:
(1) Lines 56-58, the author mentioned that “GABA treatment resulted in increased plant height and leaf area,suggesting enhanced root strength and salt tolerance”. Is there a necessary connection between the increase in plant height and leaf area and the enhancement of root strength and salt tolerance? Please state clearly.
(2) Lines 87-88, the author mentioned that there is relatively little research on GABA biosynthesis in barley. Why does this specifically refer to barley? Are there any related studies on wheat and corn?
(3) The Polyamine degradation pathway shown in Figure 2 should be annotated in the text of lines 118- 133 for better understanding of the synthesis process of this pathway.
(4) The content of Degradation of GABA in Plants in Part 3 is relatively limited. and it is recommended to integrate it with the content of Polyamine Degradation Pathway in Part 2.
(5) Line 166, “GAD (glutamate decarboxylase) genes have been identified in a variety of plants (Figure 1)", "Figure 1” should be changed to “Table 1” here. In addition, it is suggested to add relevant research on GAD genes in cereal crops such as wheat and rice in Table 1.
(6) Line 182, The author mentioned that the sentence “Overexpression of OsMYB55 in transgenic rice lines” is ambiguous and should be changed to ”Overexpression of OsMYB55 in rice lines”.
(7) Lines 186-188, “This transcription factor was shown to bind to the promoters of these genes in vitro and activate them in tobacco cells”, This statement is inaccurate. It is recommended to review the literature again.
(8) Lines 347-357, the author's description of the effect of ultrasonic treatment of germinal brown rice on GABA content is too redundant. It would be best to simplify the language.
(9) Some gene names in the article need to be italicized, such as “PopGABA-T1 “in line 198, “PopSSADH1 “in line 201, and Latin names of some species need to be italicized, such as “Arabidopsis thaliana” in line 216.
Comments on the Quality of English Language
(1) Line 130: Add the word ” respectively” following the word “synthase" in the statement:"... through reactions catalyzed by spermidine synthase and spermine synthase, respectively. ”
(2) Lines 125-133: It is recommended not to use “converted into “multiple times (4 times) in a row.
(3) In Table 1, add “S “ after the words “Author” in the statement: "Authors of the reference.”
(4) Lines 173-174: Modify this sentence “GAD3 and GAD5, on the other hand, have lower expression levels in the same tissues” to “On the contrary, GAD3 and GAD5 have lower expression levels in the same tissues.”
(5) Line 190: Replace the word “using" with “in" in the statement. “Research in Arabidopsis revealed that ...”
(6) Line 292 and line 295: Remove the words “GAD” in the statement: “in soybean...”, “In rice bran...”.
Author Response
Reviewer 1:
Comments and Suggestions for Authors
This manuscript provided a detailed overview of the biological functions, synthesis mechanisms, and regulatory pathways of GABA, as well as a detailed summary of related research on GABA in plants. Finally, key enzyme genes controlling GABA biosynthesis, regulating transporters involved in GABA metabolism, and enhancing GAD activity were proposed to promote GABA synthesis in plants. Overall, this is a comprehensive study that provides theoretical basis and research direction for crop nutritional quality and resistance breeding. However, my points should be well addressed below:
Author response: Thanks for your positive comments for our manuscript. Your comments is very important to improve our manuscript. All author‘s response as following:
- Lines 56-58, the author mentioned that “GABA treatment resulted in increased plant height and leaf area,suggesting enhanced root strength and salt tolerance”. Is there a necessary connection between the increase in plant height and leaf area and the enhancement of root strength and salt tolerance? Please state clearly.
Author response: Thank you for pointing it out. The increase in plant height and leaf area is not necessarily related to the increase in root strength and salt tolerance, and both belong to a parallel relationship. GABA treatment can increase plant height and leaf area, enhance root strength and salt tolerance.(lines 65-67)
- Lines 87-88, the author mentioned that there is relatively little research on GABA biosynthesis in barley. Why does this specifically refer to barley? Are there any related studies on wheat and corn?
Author Response: Thank you for pointing it out. Due to the limited research on GABA biosynthesis in barley, which is a major cereal crop with abundant GABA content, it is a very important research direction. In addition, our research focus on barley. However, related studies are relatively abundant in rice and corn.
- The Polyamine degradation pathway shown in Figure 2 should be annotated in the text of lines 118- 133 for better understanding of the synthesis process of this pathway.
Author response: Thank you for pointing it out. We have incorporated the relevant literature.([74-76])
- The content of Degradation of GABA in Plants in Part 3 is relatively limited. and it is recommended to integrate it with the content of Polyamine Degradation Pathway in Part 2.
Author response: Thank you for pointing it out. We have restructured the article to combine the two.(lines 262-267)
- Line 166, “GAD (glutamate decarboxylase) genes have been identified in a variety of plants (Figure 1)", "Figure 1” should be changed to “Table 1” here. In addition, it is suggested to add relevant research on GAD genes in cereal crops such as wheat and rice in Table 1.
Author response: Thank you for pointing it out. We have included the data from rice and cotton in table 1. But there is less research on wheat, which has an uncertain number of GAD genes, we don’t mention wheat. Please see table 1.
- Line 182, The author mentioned that the sentence “Overexpression of OsMYB55 in transgenic rice lines” is ambiguous and should be changed to ”Overexpression of OsMYB55 in rice lines”.
Author response: Thank you for pointing it out. The modification has been made on line 327-329.
- Lines 186-188, “This transcription factor was shown to bind to the promoters of these genes in vitro and activate them in tobacco cells”, This statement is inaccurate. It is recommended to review the literature again.
Author response: Thank you for pointing it out. The manuscript has been revised to “This transcription factor has been shown to bind to the promoters of these genes to activate them” Please see line 332-333.
- Lines 347-357, the author's description of the effect of ultrasonic treatment of germinal brown rice on GABA content is too redundant. It would be best to simplify the language.
Author response: Thank you for pointing it out. We has simplified the description of the effect of ultrasonic treatment of germinated brown rice on GABA content in this section. Please see line 472-475.
- Some gene names in the article need to be italicized, such as “PopGABA-T1 “in line 198, “PopSSADH1 “in line 201, and Latin names of some species need to be italicized, such as “Arabidopsis thaliana” in line 216.
Author response: Thank you for pointing it out. We have made the revision.
Comments on the Quality of English Language
- Line 130: Add the word ” respectively” following the word “synthase" in the statement:"... through reactions catalyzed by spermidine synthase and spermine synthase, respectively. ”
Author response: Thank you for pointing it out. It has been added on line 281.
- Lines 125-133: It is recommended not to use “converted into “multiple times (4 times) in a row.
Author response: Thank you for pointing it out. We have already corrected it on lines 276-280.
- In Table 1, add “S “ after the words “Author” in the statement: "Authors of the reference.”
Author response: Thank you for pointing it out. We had modified in Table 1.
- Lines 173-174: Modify this sentence “GAD3 and GAD5, on the other hand, have lower expression levels in the same tissues” to “On the contrary, GAD3 and GAD5 have lower expression levels in the same tissues.”
Author response: Thank you for pointing it out. We have corrected it on line 318.
- Line 190: Replace the word “using" with “in" in the statement. “Research in Arabidopsis revealed that ...”
Author response: Thank you for pointing it out. We have corrected it on line 335-336.
- Line 292 and line 295: Remove the words “GAD” in the statement: “in soybean...”, “In rice bran...”.
Author response: Thank you for pointing it out. We have deleted it.
All the corrections were marked in red font.
We look forward to hearing from you regarding our submission. We would be glad to respond to any further questions and comments that you may have
Thanks a lot!

Reviewer 2 Report
Comments and Suggestions for Authors
Overall, the study is well designed and presented in a good way, it meets the requirements for a review, the objective is in line with the information presented. Before recommending the publication of this article, some suggestions should be resolved.
Minor revisions
It is recommended to add a graphical summary
Materials and methods: What were the keywords used, inclusion criteria for the articles, etc.
I strongly recommend using the Prisma Statement flowchart for data selection. (https://www.prisma-statement.org/ )
Divide the conclusion and perspectives section
In the references section, attach the doi
Comments on the Quality of English LanguageMinor editing of English language required
Author Response
Reviewer 2:
Comments and Suggestions for Authors
Overall, the study is well designed and presented in a good way, it meets the requirements for a review, the objective is in line with the information presented. Before recommending the publication of this article, some suggestions should be resolved.
Author response: Thanks for your positive comments for our manuscript. Your comments is very important to improve our manuscript.
Minor revisions
- It is recommended to add a graphical summary
Author response: Thank you for pointing it out. We have already made a graphical summary.
Please see the graphical summary.
- Materials and methods: What were the keywords used, inclusion criteria for the articles, etc.I strongly recommend using the Prisma Statement flowchart for data selection. (https://www.prisma-statement.org/ )
Author response: Thank you for your suggestions, We have used it to make our data clearer. Please see new Materials and methods.
- In the references section, attach the doi
Author response: Thank you for your suggestions, We have added doi of all the papers with doi, and added it with hyperlinks. Please see new reference part.
- Divide the conclusion and perspectives section
Author response: Thank you for your suggestions, we have divided the conclusion and perspectives section. Please see new conclusion and perspectives section.
- Comments on the Quality of English Language
Minor editing of English language required
Author response: Thank you for your suggestions, we have made a revision in whole manuscript. All the revision were marked in red font.
We look forward to hearing from you regarding our submission. We would be glad to respond to any further questions and comments that you may have
Thanks a lot!

Reviewer 3 Report
Comments and Suggestions for Authors
The information presented in the article is generally well known. It was discussed previously in review papers several times. Some more recent data on the topic (on the role of GABA as a phytomediator, on the presence of specific GABA receptors on the surface of membranes in plant cells) are not explicitly discussed here. Overall, the manuscript is poorly structured.
Section 1.1 represents a collection of poorly systematized data on the participation of GABA in the adaptive response to stress.
Section 2.1 is called "Synthesis of GABA in plants", but the data are presented on both the biosynthesis and catabolism of GABA. It would be more appropriate to call the section "Metabolism of GABA in plants". GABA shunt is a bypass pathway of the two reactions of the TCA cycle, as a result of which GABA is formed, which is catabolized in the same pathway. A common mechanism for converting pyruvate (produced during glucose oxidation in glycolysis) is the oxidation pathway to acetyl-CoA in mitochondria with the participation of the pyruvate dehydrogenase complex. The authors presented a diagram of the pathway of pyruvate to oxaloacetate conversion via pyruvate carboxylase. This pathway occurs mostly in microorganisms. The choice of the authors in this case is unclear. Why the classic scheme pyruvate-acetyl-CoA-citrate is not considered?
Figure 1. This figure shows some kind of membrane (we can assume this is a plasma membrane, however, there are no explanations in the text or in the caption) and probably the total internal compartment of the cell. Glycolysis, the TCA cycle, GABA shunt, and the GS/GOGAT pathway have different localizations in the cell. This should be indicated in the diagram. In addition, the TCA cycle was indicated in the caption, but not in the diagram itself, indicating OAA-2OG with a line. Moreover, GDH in plants possess both NADH and NADPH specificity. It would be more correct to indicate the object, the compartment/tissue/organ, and NAD(P)/NAD(P)H. In addition, the caption includes calmodulin, which is not present in the figure. There is no description/explanation of the role of the GABA transporter and GLMTs in the text. The indicated mechanism of GABA transport through the plasma membrane, shown in the figure, is not sufficiently explained. It seems that the caption to the figure, the figure, and the text of the article are parts of different documents.
Section 3 discussed the GABA catabolism pathway through the GABA shunt. It is worth noting that the text describes the transamination of GABA into succinic semialdehyde under the action of 2-OG-dependent GABA transaminase. In this case, 2-OG acts as an acceptor of the amino group with the formation of glutamate, however, the figure shows a reaction catalyzed by pyruvate-dependent GABA transaminase (acceptor – pyruvate). Information on the number of GAD genes in maize has been revised several times since 2022. Currently, three of them are characterized. The presented data on this are too outdated; more recent data are needed.
The explanations provided in Section 4.1 regarding changes in the expression of the genes of the GABA shunt enzymes are too superficial – there is no description of what exactly was obtained and what it indicates.
Author Response
Comments and Suggestions for Authors
- The information presented in the article is generally well known. It was discussed previously in review papers several times. receptors on the surface of membranes in plant cells) are not explicitly discussed here.
Author response: Thank you for pointing this out. We have added some recent progress to further discuss the role of GABA in resisting abiotic stresses. We have systematically discussed the receptors on the cell membrane and organelle membranes. Please see new manuscript.
- Overall, the manuscript is poorly structured.
Author response: Thank you for pointing this out. We have improved the structure of this review to make the logical relationship of the paper as well as the overall framework clearer. Please see the whole review
- Section 1.1 represents a collection of poorly systematized data on the participation of GABA in the adaptive response to stress.
Author response: Thank you for pointing this out. We have incorporated systematic data on GABA involvement in abiotic stress.
- Section 2.1 is called "Synthesis of GABA in plants", but the data are presented on both the biosynthesis and catabolism of GABA. It would be more appropriate to call the section "Metabolism of GABA in plants".
Author response: Thank you for pointing this out. We have changed to "Metabolism of GABA in plants". Please see line 228.
- GABA shunt is a bypass pathway of the two reactions of the TCA cycle, as a result of which GABA is formed, which is catabolized in the same pathway. A common mechanism for converting pyruvate (produced during glucose oxidation in glycolysis) is the oxidation pathway to acetyl-CoA in mitochondria with the participation of the pyruvate dehydrogenase complex. The authors presented a diagram of the pathway of pyruvate to oxaloacetate conversion via pyruvate carboxylase. This pathway occurs mostly in microorganisms. The choice of the authors in this case is unclear. Why the classic scheme pyruvate-acetyl-CoA-citrate is not considered?
Author response: Thank you for pointing this out. We reviewed the relevant literature, in plants indeed the classic scheme pyruvate-acetyl-CoA-citrate, which we have modified. Please see line 242-246.
- Figure 1. This figure shows some kind of membrane (we can assume this is a plasma membrane, however, there are no explanations in the text or in the caption) and probably the total internal compartment of the cell. Glycolysis, the TCA cycle, GABA shunt, and the GS/GOGAT pathway have different localizations in the cell. This should be indicated in the diagram.
Author response: Thank you for pointing this out. We have optimized and modified the graph. The outermost membrane on the graph is the cell membrane, and the inner membrane inside is the mitochondrial membrane. We distinguish between different sites in the GABA metabolism. Please see figure 1.
- In addition, the TCA cycle was indicated in the caption, but not in the diagram itself, indicating OAA-2OG with a line.
Author response: Thank you for pointing this out. We added the TCA cycle to optimize the figure. Please see figure 1.
- Moreover, GDH in plants possess both NADH and NADPH specificity. It would be more correct to indicate the object, the compartment/tissue/organ, and NAD(P)/NAD(P)H.
Author response: Thank you for pointing this out. After reviewing the literature, we learned that the two enzymes that synthesize glutamate: glutamate synthase (GOGAT) and glutamate dehydrogenase (GDH) have corresponding classifications according to their different coenzymes. We have explained the specific classification and the tissue distribution of the two enzymes. Please see line 247-251, line 257-260.
- In addition, the caption includes calmodulin, which is not present in the figure.
Author response: Thank you for pointing this out. We have added this to Fig. Please see figure 1.
- There is no description/explanation of the role of the GABA transporter and GLMTs in the text.The indicated mechanism of GABA transport through the plasma membrane, shown in the figure, is not sufficiently explained. It seems that the caption to the figure, the figure, and the text of the article are parts of different documents.
Author response: Thank you for pointing this out. We have improved the article structure to introduce the GABA transporter. Please see line 122-227.
- Section 3 discussed the GABA catabolism pathway through the GABA shunt. It is worth noting that the text describes the transamination of GABA into succinic semialdehyde under the action of 2-OG-dependent GABA transaminase. In this case, 2-OG acts as an acceptor of the amino group with the formation of glutamate, however, the figure shows a reaction catalyzed by pyruvate-dependent GABA transaminase (acceptor – pyruvate)
Author response: Thank you for pointing this out. Under the catalysis of GABA transaminase, which relies on 2-OG (rather than pyruvate), GABA is transaminated to succinate half aldehyde, which we have corrected on the diagram to ensure the unification of text and figure. Please see figure 1.
- Information on the number of GAD genes in maize has been revised several times since 2022. Currently, three of them are characterized. The presented data on this are too outdated; more recent data are needed.
Author response: Thank you for pointing this out. We have reviewed a large amount of papers and obtained the conclusion that the current number of GAD genes in maize is 5. If you have the latest paper mentioning the number of GAD genes in maize, please point it out .
- The explanations provided in Section 4.1 regarding changes in the expression of the genes of the GABA shunt enzymes are too superficial – there is no description of what exactly was obtained and what it indicates.
Author response: Thank you for pointing this out. We have made some revision about what we get and what we indicate. Please see line 347-349, line 358-360. The progress in this filed is not too much. It is difficult to discuss deeply in this field.
All the corrections were marked in red font. We look forward to hearing from you regarding our submission. We would be glad to respond to any further questions and comments that you may have
Thanks a lot!

Round 2
Reviewer 3 Report
Comments and Suggestions for Authors
The authors' response regarding GAD genes is logical as the most recent articles on GAD genes contain information about the presence of five genes in maize. Still, NCBI database reveals only three of them. It is correct to indicate what was published earlier, however, changes in databases can lead to confusion and errors in research. In general, we can consider the issue resolved since the authors indicated the years of publication and the reader can double-check this information if desired.
Table 1: The species names of plant organisms should be given in Latin. The spelling of the author's names in the table should also be standardized, there are some misspellings, e.g. "SHELP B J, ZA R EI A" - why all capital letters, spaces in the second names?
Figures 2A and 2B are poorly integrated into the text. The indicated figure is mentioned at the beginning of Section 4, but no further references were found in the text.
Sentence: "A common feature of these GAD family genes is the presence of CAAT-box, TATA-box, and MYC elements in their promoters, which suggests potential regulation by these transcription factors [82]". The wording should be improved, otherwise, based on this sentence, it turns out that CAAT-box and TATA-box are transcription factors. These are regulatory elements that are part of the promoters of various genes.
In the text, it should be explained what are MYC (what kind of transcription factor) and OsMYB55.
Author Response
Comments and Suggestions for Authors
The authors' response regarding GAD genes is logical as the most recent articles on GAD genes contain information about the presence of five genes in maize. Still, NCBI database reveals only three of them. It is correct to indicate what was published earlier, however, changes in databases can lead to confusion and errors in research. In general, we can consider the issue resolved since the authors indicated the years of publication and the reader can double-check this information if desired.
Author response: Thank you for your further comments. These comments are very important to improve our manuscript. We will make response to your comments point by point as following:
- Table 1: The species names of plant organisms should be given in Latin. The spelling of the author's names in the table should also be standardized, there are some misspellings, e.g. "SHELP B J, ZA R EI A" - why all capital letters, spaces in the second names?
Author response: Thank you for your suggestions. We have rewritten the species name in Latin. What’s more, we have revised the author's names to make them standardized. Please see new table 1.
- Figures 2A and 2B are poorly integrated into the text. The indicated figure is mentioned at the beginning of Section 4, but no further references were found in the text.
Author response: Thank you for your suggestions. Figures 2A and 2B are re-integrated into the text. Please see line 299-300, line 358,372, line 450-451, line 482, 486. Line 492. You can find the relevant references in manuscript.(Figure 2A: [77-107] Figure 2B: [109-115]
- Sentence: "A common feature of these GAD family genes is the presence of CAAT-box, TATA-box, and MYC elements in their promoters, which suggests potential regulation by these transcription factors [82]". The wording should be improved, otherwise, based on this sentence, it turns out that CAAT-box and TATA-box are transcription factors. These are regulatory elements that are part of the promoters of various genes.
Author response: Thank you for your suggestions. We have revised it in line 310-313.
- In the text, it should be explained what are MYC (what kind of transcription factor) and OsMYB55.
Author response: Thank you for your suggestions. We have explained MYC on line 313-316 and OsMYB55 on line 319-322.
